# Lie Group Equivariant Convolutional Neural Network Based on Laplace Distribution

**Dengfeng Liao *** and **Guangzhong Liu**

College of Information Engineering, Shanghai Maritime University, Shanghai 201306, China;
gzhliu@shmtu.edu.cn
* Correspondence: liaodengfeng0004@stu.shmtu.edu.cn

**Abstract:** Traditional convolutional neural networks (CNNs) lack equivariance for transformations such as rotation and scaling. Consequently, they typically exhibit weak robustness when an input image undergoes generic transformations. Moreover, the complex model structure complicates the interpretation of learned low- and mid-level features. To address these issues, we introduce a Lie group equivariant convolutional neural network predicated on the Laplace distribution. This model's Lie group characteristics blend multiple mid- and low-level features in image representation, unveiling the Lie group geometry and spatial structure of the Laplace distribution function space. It efficiently computes and resists noise while capturing pertinent information between image regions and features. Additionally, it refines and formulates an equivariant convolutional network appropriate for the Lie group feature map, maximizing the utilization of the equivariant feature at each level and boosting data efficiency. Experimental validation of our methodology using three remote sensing datasets confirms its feasibility and superiority. By ensuring a high accuracy rate, it enhances data utility and interpretability, proving to be an innovative and effective approach.

**Keywords:** Laplace distribution; Lie group; equivariant convolutional network; image recognition

## 1. Introduction

"Interpretation is the process of giving explanations to Human" [1]. The deep learning model, such as the convolutional neural network (CNN), has poor interpretability in terms of the importance of features in overall decision making and what specifically improves the key factors of deep learning systems [2]. If a balance among data efficiency, accuracy, and interpretability could be achieved, interpretable models would become indispensable in certain application scenarios.

On the one hand, deep learning models often achieve higher learning accuracy than traditional machine learning. However, unexpected risks can arise in fields where interpretability is prioritized, such as healthcare and the military [3]. On the other hand, deep learning models require a vast amount of training data, while humans can grasp new concepts with a few labels. The issue of imitation poses a significant challenge in AI research [4]. Practically, enhancing the statistical efficiency of deep learning is crucial since acquiring a large volume of labeled data is costly, especially in fields like medical imaging or military remote sensing image recognition [5]. To address this problem, Gidaris et al. [6] utilized an unsupervised semantic feature learning approach, employing RotNet to train convolutional neural networks (ConvNets) for learning image features capable of identifying two-dimensional rotations applied to input images. For the ConvNet model to successfully predict an image's rotation, it must learn to locate prominent objects within the image, recognize their orientations and types, and then associate the object orientations with the original image. Experimental results indicate that the ConvNet model's performance was merely 2.4 points lower than the supervised approach. Feng et al. [7] implemented a self-supervised method to incorporate rotation invariance into the

feature learning framework. Their proposed model aimed to learn a split representation encompassing both rotation-related and rotation-independent components and train neural networks by jointly predicting image rotation and distinguishing individual instances. In contrast, this paper proposes a supervised learning method leveraging limited labeled data while exploiting latent, general symmetry in images.

In fact, symmetry, including translational symmetry, is pervasive in human vision, and is also evident in computer vision tasks. The label function and data distribution are approximately invariant to offset [8]. Traditional convolutional neural networks (CNNs) are equivariant to translation, but they do not exhibit equivariance to rotation and other generic geometric transformations. Recognizing that groups possess robust symmetrical structures, it was highlighted in Cohen's seminal paper [9] that rotation is not symmetric with the convolution operation (correlation is not an equivariant map for the rotation group). Consequently, group convolution was designed to replace the traditional convolutional layer. By doing so, the convolutional network was improved from a group theory perspective, thereby constructing the group equivariant convolutional neural network (G-CNN). This enhancement boosts the network's expressiveness without increasing the number of parameters or data augmentation, making the network equivariant to discrete rotation because group symmetry reduces sample complexity. Larocca et al. [10] used either the continuous Lie group or the discrete symmetry group to render quantum machine learning (QML) more geometric and group-theoretic.

Regrettably, the group convolutional neural network has not garnered substantial attention. One primary reason for this neglect is the inherent difficulty in computing and storing the response of each group in the natural realization of group convolution, rendering it unfit for infinite groups. To address this, Cohen et al. [11,12] proposed a broader framework: steerable CNNs. This model eschews storing feature map values in each group, opting instead to store the Fourier transform of the feature map. This method allows the extension of discrete groups to compact groups and partially continuous groups. Similarly, Xu et al. [13] introduced a unified framework for group equivariant networks on homogeneous spaces from a Fourier perspective. Their research demonstrated that when the stabilizer subgroup is a compact Lie group, the Fourier coefficients are sparse and non-zero for certain domains, providing a better characterization of the kernel space.

Despite intriguing theoretical developments and gradual improvements to the group equivariant neural network, its practical application remains limited. Cohen et al. [14] proposed spherical CNNs, which extend the convolutional network to extract features from spherical images. Additionally, they implemented the broad (non-commutative) fast Fourier transform (FFT) to calculate group convolution (cross-correlation), thereby addressing the geometric distortion issue caused by the traditional CNN's direct unfolding of spherical images. As the theory found more applications, and new scenarios emerged, Worrall et al. [15] proposed harmonic networks (H-Nets) to replace conventional CNN filters with cyclic harmonics, ensuring translation and rotation equivariance. Weiler et al. [16] demonstrated that equivariant convolution is a general linear map on $\mathbb{R}^3$. Experiments revealed the effectiveness of three-dimensional steerable CNNs for amino acid propensity prediction and protein structure classification, both exhibiting inherent SE(3) symmetry.

However, many G-CNNs are constrained to discrete groups or partially continuous compact groups. To overcome this limitation, Bekkers [17] proposed a modular framework to design and implement G-CNNs for any Lie group. This approach enabled local, irregular, and deformable convolutions in networks through local, sparse, and non-uniform B-splines. MacDonald et al. [18] proposed a framework applicable to any finite-dimensional Lie group, addressing the issue of prior group convolutional networks needing to make strong assumptions about groups, and thus being inapplicable to the affine group and the homography group. Unfortunately, most models have been tested on benchmark datasets such as MNIST and CIFAR-10. While valid conclusions can be drawn from these tests, they remain somewhat detached from practical applications, explaining the lack of extensive attention. Testing methods from the aforementioned literature on the AID remote

sensing datasets revealed increased expenses. Remote sensing, typically used in military and agricultural fields, often involves data collection at variable angles rather than a fixed one, leading to fluctuations in rotation angles, scales, and other factors in the collected images [19]. In light of this, models are trained on large quantities of disturbance data (either by collecting more data or enhancing data processing). However, this method is suboptimal. Given the above discussion, we propose combining the Laplace distribution and the Lie group with general symmetry to directly enhance the network model. This would enable its application to image recognition tasks in fields like remote sensing and healthcare, which suffer from a lack of labeled data or require interpretability, ultimately improving data utilization and enhancing interpretability.

The group equivariant convolutional neural network represents a natural generalization of the convolutional neural network. Although this deep learning methodology outperforms traditional manual feature-based methods, it is not without limitations [20]. For instance, some studies utilize regional characteristics [21] or employ mid-level information to describe image features [22]. Other models amalgamate local and global features [23], as shown in the image. However, they often overlook the correlation between image regions and the interconnections between different image features, despite the significance of these internal links.

Thus, the primary objective of this paper is to present a Lie group equivariant convolutional neural network employing regional multifeature fusion. This approach enhances data efficiency, offers interpretability from a Lie group perspective, and ensures high accuracy. The main framework, depicted in Figure 1, encompasses two primary components. The first part is the construction of Laplace Lie group feature descriptors. The Laplace distribution, being a more flexible and heavy-tailed distribution family than the Gaussian distribution, is capable of handling the analysis of continuous data with outliers [24]. Therefore, we consider the affine Lie group and its left polar decomposition based on the Laplace distribution to construct a joint representation of multiple features, yielding a multichannel, low-dimensional Lie group feature map. The second part entails the exploration of the Lie group equivariant convolutional neural network. The main contributions are as follows:

1. We leverage the Lie group of the Laplace distribution function space to construct the affine Lie group. This representation illustrates the relationship between different regions and features of the image, formulates the spatial information based on image decomposition, and preserves the geometric and algebraic structure of the pre- and post-mapping spaces, drawing upon Lie group theory.

2. We achieve multifeature joint representation through the covariance and mean of the Laplace distribution. This approach integrates low- and mid-level features and reflects correlations among different features. Moreover, the affine Lie group resulting from mapping is a $d$-dimensional real symmetric matrix Lie group, possessing advantageous computational performance and noise resistance.

3. The Lie group equivariant convolutional neural network, based on the Laplace distribution, offers excellent interpretability from a Lie group theory perspective, and significantly enhances data efficiency in terms of generalized symmetry. Its efficacy is apparent in practical remote sensing recognition experiments, positioning it as a lightweight neural network with wide-ranging application prospects.

Subsequent sections of this paper will delve into related work (Section 2), the construction and computation of the Lie group feature map (Section 3), the development and equivariant features of each level in the Lie group equivariant neural network (Section 4), and an experiment involving training on the original training set of three remote sensing datasets and testing on the enhanced data test set (Section 5). The final section will provide conclusions and prospects for future research (Section 6).

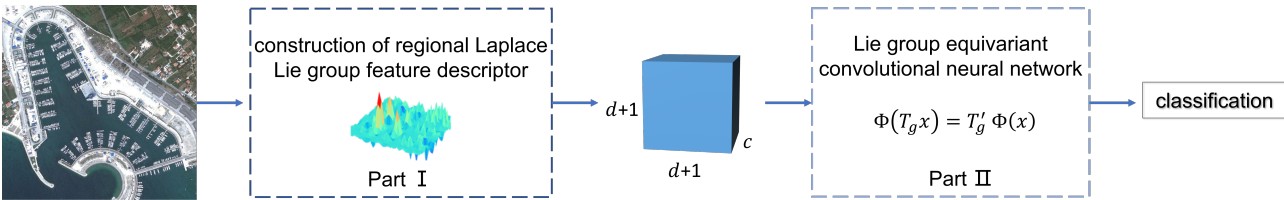

**Figure 1.** An outline of the general framework. Part I constructs the Laplace Lie group feature descriptor, where '*d*' represents the number of selected mid- and low-level features and '*c*' is the number of image decompositions. Part II describes the Lie group equivariant convolutional neural network, culminating in the final category output.

## 2. Related Work

The covariance matrix space is a Riemannian manifold, prompting the proposal of a covariance descriptor using affine, invariant Riemannian measures [25]. Maryam [26] constructed a covariance descriptor based on the feature map extracted by convolution, and utilized a support vector machine with a logarithm of a matrix kernel for image classification using the polarimetric synthetic aperture radar (PolSAR), yielding promising results. Li et al. [27] proposed the local log-Euclidean multivariate Gaussian descriptor ($L^2$EMG), demonstrating its effectiveness in image classification. The construction process of this descriptor employs the Lie group to maintain the geometric and algebraic structures within the space. The Gaussian descriptor, which uses a model to address the problem, includes the interpolated Gaussian descriptor (IGD) [28]. This descriptor represents a novel one-class classification (OCC), and a smooth Gaussian descriptor can also be learned for small or noisy samples, demonstrating superior precision and robustness.

The deep learning model related to our work includes the CapsNet [8] and the group equivariant neural network [5,9]. Reference [29] integrates the CapsNet into the CNN framework, although CapsNets have limited theoretical support regarding various invariants [30]. The execution of convolutional operations in CNNs is the summation of responses between filters and features, with filter translation proven to be equivariant. When combined with more general, generalized symmetry from group theory, Finzi et al. [31] proposed the implementation of the Monte Carlo analysis method to approximate Lie convolution, ensuring that the convolutional layer is equivariant to translation and rotation is equivariant. Similarly, Tycho et al. [32] suggested defining Lie group filters with sparse representations through an anchor point, improving parameter efficiency.

## 3. Lie Group Representation of Laplace Distribution

This section treats the Laplace distribution as the image representation and maps the distribution functions to the Lie group based on Lie group theory, acquiring the corresponding affine matrix Lie group. Further division can be achieved through the left coset. Figure 2 displays the overall mapping steps. The Lie group, acquired by isomorphic mapping, preserves the algebraic and geometric structure of the space, providing greater detail.

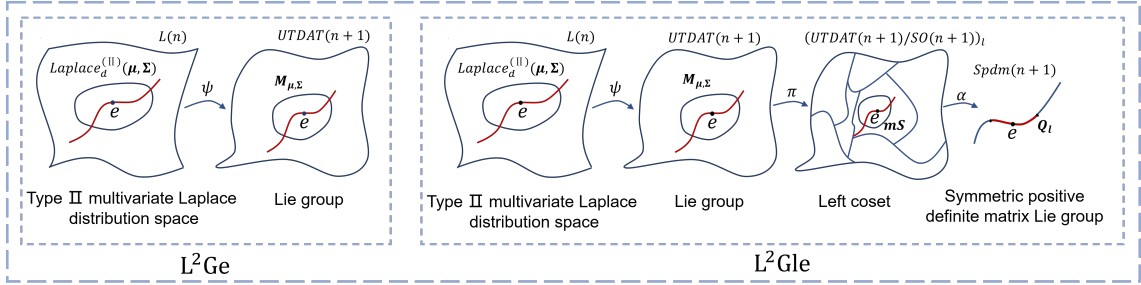

**Figure 2.** Laplace Lie group embedding and Laplace Lie group left polar decomposition embedding methods.

### 3.1. Construction of the Laplace Feature Map

Let $I$ represent an input original image, and $F(x,y)$ denote the $d$-dimensional low-level or mid-level feature vectors extracted from the image:

$$F(x,y) = \phi(I,x,y). \tag{1}$$

where $\phi$ symbolizes the mapping function and the pixel points $(x,y)$ in the image map to a $d$-direction vector. We select one region $R \in \mathbb{R}^{w \times h}$ and extract a group of $w \times h$ regional features $\left\{ \mathbf{x}_i \in \mathbb{R}^{d \times 1}, i = 1, \ldots, w \times h \right\}$ based on Equation (1). Through likelihood statistics, this region can be expressed via the II multivariate Laplace distribution [24] with the following parameters:

$$E(\mathbf{x}) = \boldsymbol{\mu}, \text{ and } Var(\mathbf{x}) = \frac{\pi}{2}\boldsymbol{\Sigma} + \left(2 - \frac{\pi}{2}\right)\text{diag}(\boldsymbol{\Sigma}). \tag{2}$$

Therefore, $\mathbf{x}$ adheres to the type II multivariate Laplace distribution, denoted as $\mathbf{x}_i \sim Laplace_d^{(II)}(\boldsymbol{\mu}, \boldsymbol{\Sigma})$. $\boldsymbol{\mu}$ is the location/mean vector parameter, $\boldsymbol{\Sigma} = (\sigma_{ii})$ is the scale parameter matrix. Its formula is:

$$\hat{\boldsymbol{\mu}}^M = \frac{1}{n}\sum_{j=1}^{n} x_j, \text{ and } \hat{\boldsymbol{\Sigma}}^M = \left(\hat{\sigma}_{ii'}^M\right). \tag{3}$$

where

$$\hat{\sigma}_{ii}^M = \sum_{j=1}^{n} \frac{(x_{ij} - \overline{x}_i)^2}{2(n-1)}, i = 1, \ldots, d,$$

$$\hat{\sigma}_{il'}^M = 2\sum_{j=1}^{n} \frac{(x_{ij} - \overline{xi}_i)(x_{kj} - \overline{x}_k)}{(n-1)\pi}, i, i' = 1, \ldots, d,$$

and $i \neq i', \overline{x}_i = \sum_{j=1}^{n} x_{ij}/n$.

Let's consider a space formed by a type II multivariate Laplace distribution function, denoted as $L(n)$. Similar to the Gaussian space being a Riemannian space, if $\mathbf{x}_0 \sim Laplace_d^{(II)}(\boldsymbol{\mu}_0, \boldsymbol{\Sigma}_0)$, the affine transformation $\mathbf{x}_k = \boldsymbol{P}\mathbf{x}_0 + \boldsymbol{\mu}$ exists, it is clear that $\mathbf{x}_k$ is a random variable that conforms to the type II multivariate Laplace distribution, i.e., $\mathbf{x}_k \sim Laplace_d^{(II)}(\boldsymbol{\mu}_k, \boldsymbol{\Sigma}_k)$, where $\boldsymbol{\mu}$ is the location/mean vector parameter and $\boldsymbol{P}$ is the matrix of scale/covariance matrix decomposition, i.e., $\boldsymbol{\Sigma} = \boldsymbol{P}\boldsymbol{P}^T$. In $L(n)$, any random sample of type II multivariate Laplace distribution can undergo the following affine transformation:

$$\psi : L(n) \rightarrow L(n), \\ \psi(\mathbf{x}_{k_0}) = \mathbf{x}_k, \mathbf{x}_k = \boldsymbol{P}\mathbf{x}_{k_0} + \boldsymbol{\mu}, \boldsymbol{\Sigma} = \boldsymbol{P}\boldsymbol{P}^T. \tag{4}$$

However, Equation (4) is not the only mapping, since $\boldsymbol{\Sigma} = \boldsymbol{P}\boldsymbol{P}^T$ has multiple solutions. It is important to note that the decomposition is restricted to Cholesky decomposition; the solution $\boldsymbol{P}$ is the only upper triangular matrix. With the use of this decomposition, the covariance $\boldsymbol{\Sigma}$ of the type II multivariate Laplace distribution is presumed to be a positive definite (symmetric) matrix, making $\boldsymbol{\Sigma}^{-1} = \boldsymbol{L}\boldsymbol{L}^T$ and $\boldsymbol{L}^T$ the upper triangular matrix directly, i.e., $\boldsymbol{\Sigma} = \boldsymbol{L}^{-T}\boldsymbol{L}^{-1} = \boldsymbol{P}\boldsymbol{P}^T$, where $\boldsymbol{P}$ is an invertible positive definite upper triangular matrix. The computational cost for calculating the full covariance matrix $\boldsymbol{\Sigma}$ is $O(n^3/3)$. In Equation (4), $\mathbf{x}_k \sim Laplace_d^{(II)}(\boldsymbol{P}\mathbf{x}_{k_0} + \boldsymbol{\mu}, \boldsymbol{P}\boldsymbol{P}^T)$ and the affine mapping $\psi$ are mapped one-to-one. Its matrix form of mapping is expressed as follows:

$$\begin{bmatrix} \mathbf{X}_k \\ 1 \end{bmatrix} = \begin{bmatrix} \boldsymbol{P} & \boldsymbol{\mu} \\ \mathbf{0}^T & 1 \end{bmatrix} \begin{bmatrix} \mathbf{X}_{k_0} \\ 1 \end{bmatrix}, \tag{5}$$

and

$$M_{\mu,\Sigma} = \begin{bmatrix} P & \mu \\ 0^\mathsf{T} & 1 \end{bmatrix} \tag{6}$$

belongs to the upper triangular definite affine transform [33], $M_{\mu,\Sigma} \in UTDAT(n+1))$.

According to Lie group theory, reversible $UTDAT(n+1)$ is the Lie subgroup of $Gl(n, \mathbb{R})$, where matrix multiplication and inversion correspond to the addition and inversion of groups, respectively. It is clear that $UTDAT(n+1)$ is closed under matrix multiplication and inversion. In accordance with the aforementioned affine mapping, the type II multivariate Laplace distribution function space is equivalent to the space where matrix $UTDAT(n+1)$ is located, and it is the Lie group manifold space. Using the Cholesky decomposition algorithm, each element of $P$ can be written as a combination of elementary arithmetic or square arithmetic of elements in $\Sigma^{-1}$. As Cholesky decomposition is differentiable and unique, a Lie group isomorphic relation exists between $L(n)$ and $UTDAT(n+1)$.

Naturally, the space of the Laplace distribution function is mapped to the Lie group space:

$$\text{Laplace}_d^{(\mathrm{II})}(\mu, \Sigma) \overset{\psi}{\Rightarrow} M_{\mu,\Sigma} \in UTDAT(n+1). \tag{7}$$

We denote $M$ as the class I feature matrix of the Laplace Lie group.

Apart from the direct transformation of Equation (7), we can seek more detailed embedding methods that better reflect the structure of $L(n)$ space based on equivalence partitioning of groups. The objective of partitioning is to study group structures. The so-called division involves partitioning a set into several disjoint union sets [34]. For example, the even number is the subgroup of $\mathbb{Z}^+$ and the odd numbers are derived from even number translation (+1). Such a structure reflects its translational invariance. Apart from this integer group $\mathbb{Z}^+$ based on the remainder mod of $n$, i.e., congruence classes in number theory. Division is not unique, so appropriate and reasonable divisions may be considered to reflect the structure of the group being studied. This allows for the achievement of ideal properties such as translation invariance and rotational invariance on the image.

According to discussions on cosets in Section 1.7 of Reference [35] and Section 5.2 of Reference [36], one coset $UTDAT(n+1)/SO(n+1)$ can be found on $UTDAT(n+1)$, where $SO(n+1)$ is the orthogonal matrix group with a determinant of 1. Generally, the Lie group has an analytic mapping based on the discussion:

$$\begin{cases} \pi : UTDAT(n+1) \to (UTDAT(n+1)/SO(n+1))_l, \\ \pi(M) = mS, \ mS = \{mSO | m \in UTDAT(n+1)\}, \end{cases} \tag{8}$$

where the symbol $mS$ only represents the left coset obtained by the quotient mapping.

Based on matrix polar coordinate decomposition [37], the following mapping shall be constructed:

$$\begin{cases} \alpha : (UTDAT(n+1)/SO(n+1))_l \to Spdm(n+1)_l, \\ \alpha(mS) = S_l, \end{cases} \tag{9}$$

wherein $Spdm(n+1)$ refers to the symmetric positive-definite matrix group and $M_{\mu,\Sigma} = S_l R$ is the only left polar decomposition of the matrix $M_{\mu,\Sigma}$. $R$ is the adjugate matrix of $S_l$, and $R$ is the closest to $M$:

$$R = \arg \min_{O \in O(n+1)} \| M_{\mu,\Sigma} - O \|_F, \tag{10}$$

wherein $O(n+1)$ is the dimensional orthogonal group of $n+1$, and $S_l = \left(M_{\mu,\Sigma} M_{\mu,\Sigma}^T\right)^{1/2}$, namely,

$$S_l = \begin{bmatrix} \Sigma + \mu\mu^T & \mu \\ \mu^T & 1 \end{bmatrix}^{\frac{1}{2}}. \tag{11}$$

Furthermore, the eigenvalue of $S_l$ is equal to the singular value of $M_{\mu,\Sigma}$.

Thus, the left polar decomposition equivalent partition method is as follows:

$$\text{Laplace}_d^{(\text{II})}(\boldsymbol{\mu}, \boldsymbol{\Sigma}) \overset{\psi}{\Rightarrow} M_{\boldsymbol{\mu},\boldsymbol{\Sigma}} \in UTDAT(n+1) \overset{\pi}{\Rightarrow} mS \overset{\alpha}{\Rightarrow} S_l. \tag{12}$$

We denote $S_l$ as the class II feature matrix of the Laplace Lie group. Additionally, $S_r$ can be obtained through the right polar decomposition.

The Lie group is undoubtedly a vital and unique differential manifold. The first class of Lie group feature map, which is derived from the distribution function, is an upper triangular definite affine transform. The second class is a real, symmetric, positive-definite matrix. Regardless of the original image size, the feature map information encapsulates the correlation between different low-level (middle-level) features, offering efficient computation and substantial noise immunity by implementing average filtering on high-noise samples.

### 3.2. Calculation of the Laplace Lie Group Feature Matrix

As per Equation (1), in [27], a total of 14 combinations of changes in different low-level features are discussed, and the Gaussian distribution features are constructed using a set of 17 dimensional raw features, including grayscale value, five types of the first order and three types of the second order. Reference [20] adopts a set of 14 dimensional covariance feature descriptors, including position, RGB, YCbCr, first-order degree, second-order degree, Gabor, and LBP. To overcome the information deficiency in the basic original features, prone to be affected by the visual angle and size, one can adopt the standardized 18 features listed below:

$$\begin{aligned} F(x,y) = [&gray, N_r, N_g, N_b, C_b, C_r, \\ &Raberts_x, Ralerts_y, Scharr_x, Scharr_y, \\ &Prewitt_x, Prewitt_y, Kirsh_{max}, Sobel_{2x}, Sobel_2y, \\ &Laplace_{sum}, LBP_{(x,y)}, Gabor_{(x,y)}], \end{aligned} \tag{13}$$

where *Roberts*, *Scharr*, and other operators are utilized in tandem to find edge features. The symbol $LBP_{(x,y)}$ denotes that it is within the window of $3 \times 3$, where a binary operation is performed at eight pixel points to derive the LBP values. This operator, which describes the local texture image, offers rotational and grayscale invariance. It signifies that it applies the Gabor filter to $Gabor_{(x,y)}$, whose frequency and direction mirror those of the human visual system, thereby ensuring better direction and scale selection.

To discern differences between image areas and further scrutinize item-specific local features, Bouteldja et al. [38] divided high-resolution images into 17 subregions and extracted four-dimensional feature vectors, effectively counteracting issues of illumination, scale, and displacement. Xu et al. [20] divided remote sensing images into 11 blocks and employed 14-dimensional features. In this study, we adopt a decomposition method that segments the image into 15 subregions, including the image's global region. Experiments revealed that, although decomposition into 15 blocks takes about 420 milliseconds longer than decomposition into 11 blocks, the accuracy improves by roughly 6.8%. Consequently, we assert that excessive decomposition regions can lead to decreased computational efficiency, while insufficient divisions may cause the loss of region-specific correlation information. Therefore, for all experimental comparisons in this study, we employ the decomposition method illustrated in Figure 3.

Consequently, the parameters in Equation (2) can be estimated across 15 regions to obtain the feature map of $15 \times (d+1) \times (d+1)$. Each path of this feature map contains correlation information between the features of a certain image region as well as feature mean information. This Lie group array matrix, based on the type II multivariate Laplace distribution, is characterized by a flexible heavy tail and better addresses the analysis of continuous data with outliers.

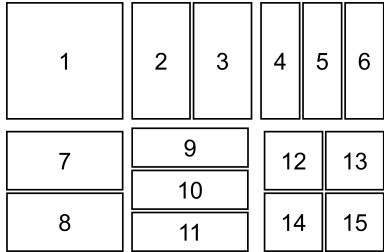

**Figure 3.** Picture decomposition. The Laplace Lie group feature matrixes are calculated in 15 area images.

## 4. Lie Group Equivariant Convolutional Neural Network

For representational space, structural information can be obtained from other connected representational spaces. Thus, the function $\Phi$ is equivariant under the transformation group $G$ and the domain $X$. This holds true if, and only if,

$$\Phi(T_g x) = T'_g \Phi(x), \tag{14}$$

where $x \in X$, $T_g, T'_g \in G$. This suggests that the essence of equivariance refers to the commutative property between operators and functions. Specifically, $T_g$ and $T'_g$ are not necessarily identical, but their transformation effects should be the same. For two transformations, $g,h$, $T(gh) = T(g)T(h)$. As inferred from (14), invariance is a specific form of invariance, with the caveat that it cannot ascertain whether the feature exists in the correct spatial configuration. Naturally, equivariance is favored over invariance [9].

To attain equivariance at every model layer, the first layer of the group equivariant convolutional neural network and its variant is used to elevate the feature map defined on the group, and thus is also known as the lifting layer. This allows group convolution and group activation to be performed on the group. Given that the multifeature joint representation discussed in Section 3 already exists on Lie groups, we need to explore how to refine it as the model that acts on the Lie group.

### 4.1. Convolutional Layer of Lie Group

To implement the autocorrelation (convolution) operation on compact or continuous groups, we define the following as per Reference [18] .

**Definition 1.** *Let the feature mapping be defined as $f \in C(G; \mathbb{R}^K)$, and the filter function as $\phi \in C_c(G; \mathbb{R}^{K \times L})$. For $u, v \in G$, define autocorrelation (convolution) as*

$$f \star \phi(u) := \int_G f(v) \cdot \phi(v^{-1}u) d\mu_L(v) \tag{15a}$$

$$= \int_G f(uv^{-1}) \cdot \phi(v) d\mu_R(v), \tag{15b}$$

*wherein $\cdot$ is the matrix multiplication, $\mu_L$ ($\mu_R$) is the left (right) Haar measure of the Lie group, $f \star \phi$ is the continuous function on $G$, and $L_v(f) \star \phi = L_v(f \star \phi)$.*

The challenge lies in identifying a method to implement the aforementioned autocorrelation (convolution) operation. The group equivariant convolutional neural network theory presented in [9,39] cannot be applied to the continuous Lie group established in Section 3. Reference [31] uses the Monte Carlo analysis method to approximate the convolution, but this approach relies on two assumptions, one of which states that Haar measures can easily be reduced to known measures of known sets to obtain the Monte Carlo estimate of (15). However, the Lie group proposed in Section 3 fails to meet this condition. In this paper, we present an approach to circumventing this assumption.

**Theorem 1.** *(Refer to [40], Section 1.2, Theorem 5). Let $\mathfrak{g}$ be the Lie algebra of the finite-dimension Lie group G, and apply $t \mapsto \xi(t)$ as the curve of $\mathfrak{g}$. Then,*

$$\left.\frac{\mathrm{d}}{dt}\right|_{t=0} \exp(X(t)) = dL_{\exp(X(0))} \frac{1 - e^{-ad_{X(0)}}}{ad_{X(0)}} \left.\frac{\mathrm{d}}{dt}\right|_{t=0} X(t), \tag{16}$$

*where $X \in \mathfrak{g}$, $ad_X : \mathfrak{g} \to \mathfrak{g}$ is the linear mapping defined by $d_X(Y) := [X, Y]$, and $dL_{\exp(X)}$ refers to the premultiplication derivative of $\exp(X) \in G$. Moreover,*

$$\frac{1 - e^{-ad_X}}{ad_X} = \sum_{k=0}^{\infty} \frac{(-1)^k}{(k+1)!} (ad_X)^k$$

*is the power series in linear map $ad_X$.*

**Theorem 2.** *(Refer to [18], Theorem 4.4). If f is an upper integrable function of G and it is 0 outside a sufficiently small neighborhood of the identity element, the Haar measurement $d_{\mu_R}$ can be found, so that*

$$\int_G f(u) d_{\mu_R}(u) = \int_{\mathfrak{g}} f(\exp(\xi)) \det\left(\frac{1 - e^{-ad - \xi}}{ad - \xi}\right) d\xi, \tag{17}$$

*wherein $d\xi$ represents the Euclidean element of the vector space $\mathfrak{g}$ ; $ad_X : \mathfrak{g} \to \mathfrak{g}$ and $(1 - e^{-ad_{-\xi}})/ad_{-\xi}$ are given in Theorem 1.*

Given that $\phi$ is defined in a sufficiently small neighborhood of the identity element in combination with Theorem 2 and Definition 1, $\widetilde{\phi} := \phi. \exp^{-1}$ is well defined. Thus, the autocorrelation (convolution) can be denoted as

$$\int_{\mathfrak{g}} f(u \exp(-\xi)) \cdot \tilde{\phi}(\xi) \det\left(\frac{1 - e^{-ad} - \xi}{ad - \xi}\right) d\xi. \tag{18}$$

According to (18), the Markov chain Monte Carlo (MCMC) method can be utilized to sample any Lie group from the Haar measure. In fact, since the Lie algebra of any Lie group is the tangent space (Euclidean space) at the identity element, we can consider

$$\xi \mapsto \det\left(\frac{1 - e^{-ad - \xi}}{ad_{-\xi}}\right)$$

as a density function. Therefore, we can apply the standard MCMC method to generate a Monte Carlo estimate of $f \star \phi$ by sampling from the Haar measure:

$$f \star \phi(u) \approx \frac{1}{N} \sum_{\xi_i \sim \exp^* \mu_R} f(u \exp(-\xi_i)) \cdot \tilde{\phi}(\xi_i). \tag{19}$$

As an illustration, we use the rotation group $C_8$ and visualize the features of traditional convolution and the Lie group convolution in the above method after a one-layer convolution operation. As shown in Figures 4 and 5, it can be seen that the features of the input rotation image vary after traditional convolution, whereas the features after Lie group convolution remain nearly constant. A potential explanation for this minor discrepancy could be the presence of equivariance-disrupting operations in the model, such as the border-filling operation performed prior to convolution.

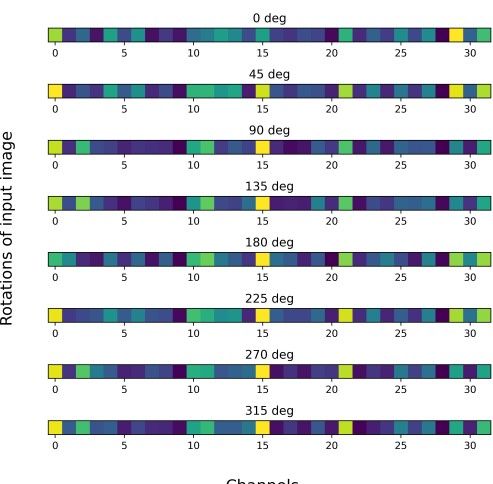

**Figure 4.** Feature-based visualization after traditional convolution.

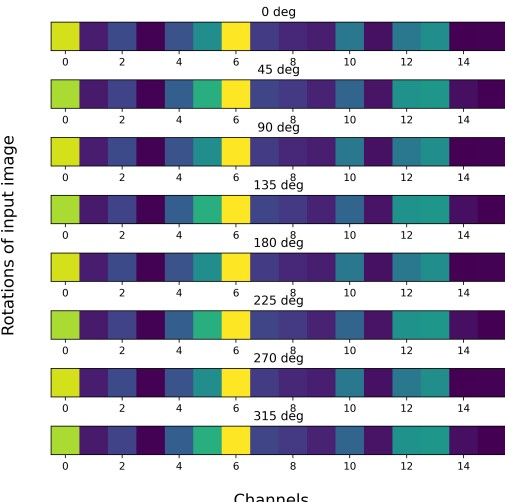

**Figure 5.** Feature-based visualization after Lie group convolution.

### 4.2. Activation Layer of Other Lie Groups

Without compromising the equivariant property, we can directly apply the nonlinear layer [9,18]. We take the feature mapping $f$ as the continuous function on $G$ and apply a nonlinear effect $\gamma : \mathbb{R}^K \to \mathbb{R}^L$, i.e., the composite operator. Thus,

$$C_\gamma f(g) = [\gamma \circ f](g) = \gamma(f(g)), \tag{20}$$

where the left transformation operator $L$ acts via recombination, allowing for $C$ to be exchanged with $L$:

$$C_\gamma L_h f = \gamma \circ \left[ f \circ h^{-1} \right] = [\gamma \circ f] \circ h^{-1} = L_h C_\gamma f, \tag{21}$$

i.e.,

$$\gamma \circ L_h(f) = L_h(\gamma \circ f), h \in G. \tag{22}$$

Therefore, the traditional nonlinear layer can be directly applied after any convolutional layer.

In [9], the design for discrete groups involves subgroup pooling and coset pooling. However, our Lie group feature map is continuous. Generally, it lacks boundaries, thereby eliminating the need for a global pooling definition. Yet, the maximum value on any compact set is well defined. Assuming a classification problem $N$, the continuous function

$f \in N(G; \mathbb{R}^K)$, along with the matrix $A \in \mathbb{R}^{K \times N}$ and the offset $b \in \mathbb{R}^N$, are provided. Furthermore, the component $f \mapsto \max_u(Af(u) + b)$ is well defined for any compact subgroup $K$ on $G$. Hence, this operation is also equivariant.

## 5. Experiment and Analysis

To assess the efficacy of the method proposed in this paper and improvements in data efficiency, we have selected three public remote sensing datasets (AID [41], NWPU-RESISC45 [42], MASATI-v2 [43]) for experimentation. The AID dataset contains 10,000 images across 30 categories. Training rates are set at 20% and 50%. The NWPU-RESISC45 dataset encompasses 45 categories of images, extracting 42 images per category, forming a test set of 1890 images. The training set contains 29,610 images, with training rates set at 10% and 20%. The MASATI-v2 dataset comprises 7389 images over seven categories (Ship, Detail, Multi, coast&ship, Sea, Coast, and Land). As the categories Ship, Detail, Multi, and Sea bear significant resemblance, we have selected only Detail, Sea, Coast, and Land as the experiment objects.

This paper strives to address issues of data scarcity in certain areas, limited interpretability of deep learning models, and high computational resource demands of variable convolutional models, such as Lie groups, which have yet to see broad application in high-resolution image recognition. A multifeature integrated Lie group equivariant convolutional neural network based on Laplace distribution is proposed, with the goal of improving interpretability through group theory and enhancing data efficiency via generalized symmetry, all while maintaining robust accuracy. As such, it is not deemed necessary to compare it with the deep learning model for remote sensing image recognition with the highest recent accuracy rate. Instead, we benchmark against a convolutional model possessing the same parameter quantity and depth as our model, comparing it with CapsNets. As illustrated in Figure 6, the model is trained on raw, untrained training data and tested on a set that has undergone random rotation, stretching, mirroring, and other transformations to ascertain its efficacy and degree of data efficiency improvement.

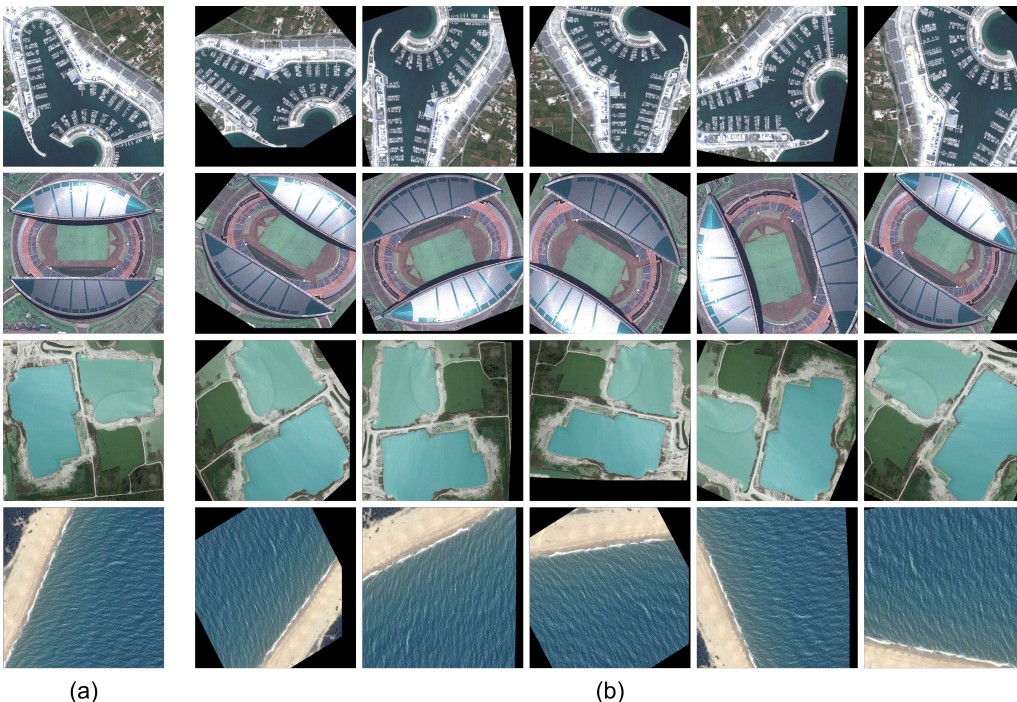

(a)                    (b)

**Figure 6.** Remote sensing image samples: Group (**a**) represents the master drawing of the original training set, while Group (**b**) denotes the master drawing of the training set following random enhancement.

Based on Sections 3 and 4, two Lie groups, namely, the directly embedded equivariant convolutional neural network ($L^2$Ge-CNN) and the left-coset-embedded equivariant convolutional neural network ($L^2$Gle-CNN) based on Laplace distribution, were designed. The main structure of the convolutional model comprises multiple modules and clustering, with each model incorporating the convolutional layer, standard layer, and equivariant linear layer of the Lie group. This experiment employs three sets of dual-module series pooling layers as the basic structure. The epochs and batch size are set to 100 and 32, respectively, while other hyperparameters utilize default values.

(1) Results of the AID dataset. The AID dataset, comprising $600 \times 600$ pixel images, is multisource, offering more challenges than single-source images. Table 1 displays a comparative analysis of the overall accuracy between different methods and the one proposed in this paper under varied training rates. A notable finding is the superior performance of the features of two embedded Lie groups over the benchmark CNN model. This can be attributed to the conventional CNN's lack of equivariance in rotation, stretching, and other transformations. Consequently, a model trained on the original dataset struggles to accurately identify the enhanced test dataset. Despite CapsNet's original design to bolster data efficiency and achieve better representation from smaller datasets, our experimental results reveal that its accuracy rate falls 10% short of the GE CapsNet, and is 17% lower than our method. Although the GE CapsNet incorporates a framework to ensure equivalency and invariance, and thus improved accuracy in our experiment, our model still achieves 4.34% higher accuracy at a 20% training rate than the GE CapsNet at a 50% training rate. This is because while CapsNet performs better than the MNIST data, it cannot be extended to a deeper structure, and its efficiency with high-resolution remote sensing images remains weak. In addition, we compare the Lie group convolutional neural network of the covariance-based feature map with the Gaussian-based feature group. The experimental comparison reveals that the features of the Gaussian Lie group outperform those of the covariance by 0.83% at a 20% training rate and by 2.75% at a 50% training rate. We found that mean information plays a less significant role as training data increases substantially, with the volume of data playing a pivotal role instead. Mean information might be diminished during convolution and pooling processes, yet it remains a vital information carrier. This is particularly true as our dimension is too low to risk substantial information loss. Lastly, we determined that the accuracy rates of $L^2$Ge-CNN and GDe+LGCNN are nearly identical at a 50% training rate, suggesting that the Gaussian feature indeed offers a good representation. However, a more finely divided $L^2$Gle-CNN model performs slightly better, by 0.19%. At a 20% training rate, the accuracy rate increases by 2%, suggesting that when data volume is small, the Laplace distribution is better equipped to handle statistical data processing for datasets like AID compared to the Gaussian distribution.

**Table 1.** Comparison of overall accuracy (OA%) of different methods on the AID dataset at training rates of 20% and 50%.

| Method | Training Rate | |
|---|---|---|
| | **20%** | **50%** |
| CNN-baseline [29] | 62.21 | 65.49 |
| CapsNet [29] | 72.56 | 75.55 |
| GE CapsNet [30] | 82.95 | 86.26 |
| Cov [20] + LGCNN | 88.32 | 90.22 |
| GDe [27] + LGCNN | 89.15 | 92.97 |
| $L^2$Ge-CNN | 90.60 | 92.45 |
| $L^2$Gle-CNN | 91.15 | 93.16 |

(2) Results of the NWPU-RESISC45 dataset. The NWPU-RESISC45 dataset, comprising $256 \times 256$ pixel images, is a remote dataset applicable to more categories than the AID dataset. As presented in Table 2, the results mirror those of the AID dataset. Overall, while the benchmark AID dataset improves, the accuracy of other models significantly decreases.

This may be due to the NWPU-RESISC45 dataset having more categories, higher intra-class diversity, and richer images. Although the benchmark CNN model performs better compared to the previous experiment, the accuracy rate falls below 70%. At a 10% training rate, the Gaussian feature sees a 3.83% increase compared to the covariance features, and a 3.08% increase at a 20% training rate. This suggests that in experimental analysis, mean information is indeed indispensable. Regardless of the training rate, the Gaussian Lie group features, as per Reference [27], achieve a similar accuracy rate on the dataset to the directly embedded Laplace Lie group features we proposed. However, the accuracy rate of the Laplace Lie group, based on the left polar decomposition, is 1% higher.

**Table 2.** Comparison of overall accuracy rate (OA%) of different methods on NWPU-RESISC45 at training rates of 10% and 20%.

| Method | Training Rate | |
|---|---|---|
| | **10%** | **20%** |
| CNN-baseline [29] | 65.09 | 68.71 |
| CapsNet [29] | 71.95 | 74.95 |
| GE CapsNet [30] | 79.88 | 82.09 |
| Cov [20] + LGCNN | 83.32 | 86.03 |
| GDe [27] + LGCNN | 87.15 | 89.11 |
| $L^2$Ge-CNN | 87.60 | 89.47 |
| $L^2$Gle-CNN | 89.15 | 90.16 |

(3) Results of the MASATI-v2 dataset. Table 3 illustrates the performance impacts of various methods on the MASATI-v2 dataset. For the experiment, only four categories were selected from the dataset, each bearing high similarity. They are all marine-related remote sensing images, thereby posing significant challenges to the model. As per the results, the $L^2$Gle-CNNmodel proposed exhibits the most effective outcomes. Both the models based on Gaussian features and covariance features see a rise in the overall accuracy rate of 2.76% and 2.96%, respectively. Regardless of the features used for embedding, the performance of the Lie group convolutional neural network constructed surpasses that of the benchmark CNN and shows a 3–6% improvement over the CapsNet-based method.

**Table 3.** Comparison of overall accuracy (OA%) of different methods on the MASATI-v2 dataset.

| Method | OA% |
|---|---|
| CNN-baseline [29] | 67.09 |
| CapsNet [29] | 75.94 |
| GE CapsNet [30] | 83.69 |
| Cov [20] + LGCNN | 86.99 |
| GDe [27] + LGCNN | 87.19 |
| $L^2$Ge-CNN | 87.97 |
| $L^2$Gle-CNN | 89.95 |

The experiments conducted on the aforementioned three datasets furnish robust evidence supporting the efficacy of our proposed approach. Utilizing Laplace-distributed Lie group features exhibits better resilience against noise. By integrating this with an equivariant Lie group convolutional neural network, the geometric and algebraic structures of the Lie group are adequately preserved. As a result, our method demonstrates superior accuracy and data utilization rates in comparison to alternate methodologies.

## 6. Conclusions

In this paper, we construct a Laplace Lie group feature map, leveraging the flexibility and heavy tail properties of the Laplace distribution as well as the group structure and differentiability strengths of the Lie group. This feature map integrates multiple low-

level or middle-level features, boasting excellent computability and information carrying capacity. Furthermore, we couple the Lie group feature map with the group equivariant convolutional neural network to propose a Lie group equivariant convolutional network model suitable for remote sensing image recognition. The construction process of the Lie group feature map is entirely rooted in the isomorphic mapping of the Laplace distribution to Lie groups and the bijective mapping between Lie groups, thereby fully respecting the geometric and algebraic structure of the Lie group. This model's implementation also theoretically guarantees the invariance of feature maps, providing interpretability from a Lie group theory perspective and validating the feasibility of the methods through three remote sensing datasets. Collectively, our method enhances data efficiency and the interpretability of deep learning while ensuring accuracy, thus offering a practical, lightweight model method for remote sensing image recognition.

In future research, improvements could be made in the following aspects: I. The construction of the Laplace Lie group feature map could be decoupled, allowing for front and rear communication design after embedding the feature map in the extracted features of the convolutional layer. II. The mixed Laplace distribution possesses better generalization capabilities and stronger characterization abilities, yet it introduces more complex commuting issues, warranting further investigation. III. Consideration should be given to extending the Lie group to the group equivariant graph neural network. Given the close connection between figures and groups, efforts could be made to enhance neural networks based on algebraic graph theory.

**Author Contributions:** Conceptualization, D.L. and G.L.; methodology, D.L.; software, D.L.; validation, D.L. and G.L.; formal analysis, D.L.; investigation, D.L.; resources, D.L.; data curation, D.L.; writing—original draft preparation, D.L.; writing—review and editing, D.L. and G.L.; visualization, D.L.; supervision, G.L.; project administration, D.L. and G.L.; funding acquisition, G.L. All authors have read and agreed to the published version of the manuscript.

**Funding:** This research received no external funding.

**Data Availability Statement:** The data presented in this study are available on request from the corresponding author.

**Conflicts of Interest:** The authors declare no conflict of interest.

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
