# Peer review of "Lie Group Equivariant Convolutional Neural Network Based on Laplace Distribution"

_remotesensing, doi:10.3390/rs15153758_

Round 1
Reviewer 1 Report
In this paper, the authors employ a Lie group feature engineering technique to enhance features, thereby enabling more sophisticated relationships to be established among objects within an image. Given the intricate nature of remote sensing data, often replete with cluttered information, the authors describe their framework as a lightweight CNN that augments the performance and robustness to noise of conventional CNNs. The authors aimed for each layer in their CNN model to exhibit equivariance, boosting model performance, interpretability, and training speed. Their model also demonstrated superior performance in situations of limited data.
General comments:
The authors provided a robust mathematical foundation; however, their results and discussions inadequately support their claim regarding the improvements attributed to their Laplace Lie group feature map. This aspect necessitates further exploration, particularly concerning enhanced model performance and interpretability. By nature, CNN offers substantial visualization capabilities to comprehend the function of each layer and their contribution to the final result. Techniques such as GradCam could significantly enhance the paper's visualization, helping readers understand how their feature engineering contributes to the model's improvements.
The authors did not fully explain how the interplay between the frontend and backend (transitioning from one feature space to another) influences the performance of their framework (what is the computational cost of that). This aspect demands a more comprehensive explanation.
The literature review could be more expansive. I expected to find discussions on self/semi-supervised learning (see RotNet paper for a basic example) and their comparison with the current model.
While the application is within Remote Sensing, I found no explicit explanation of why the proposed method is tailored to remote sensing data. Why was this specific area chosen?
The paper's writing style needs improvement and should undergo another round of review. The authors should aim for consistent passive voice throughout the manuscript.
Specific comments:
Line 115-116: Please provide an explanation or citation for this statement.
Line 258-259: The effects of these sub-regions require clear elucidation. Alternatively, justify the selection of these regions and their relevance to remote sensing images.
Line 341 and 344: Could you clarify what "The training rate is set as 20% and 50%." means?
Line 367-368: Could you provide specific reasons for choosing these values?
These are some exmaples:
Line 273-274: Could you clarify the meaning of this sentence, "the invariance is a special form of invariance and the feature of the invariance is"?
Line 288-289: "We need to find the method to realize the above autocorrelation (convolution) operation." Why the language is changed to active from passive? the authors need to follow a consistent writing workflow.
Author Response
Dear reviewer,
Thank you very much for your comments and professional advice. These opinions help to improve academic rigor of our article. Based on your suggestion and request, we have made corrected modifications on the revised manuscript.
In the revised version, the sections highlighted in red represent modifications addressing the suggestions from Reviewer 1, while the blue sections reflect changes made in response to Reviewer 2's comments. The teal highlights indicate revisions made in terms of language and writing style.
Furthermore, we would like to show the details as follows in the attachment.

Reviewer 2 Report
In this paper, the authors propose the Lie group equavariant convolutional network based on the Laplace distribution for having the computing efficiency and the ability to resisst the noise.
I read it well and there are no major corrections.
In line 47, There is a typo. "gorup"
Author Response
Dear reviewer,
Thank you very much for your comments and professional advice. These opinions help to improve academic rigor of our article. Based on your suggestion and request, we have made corrected modifications on the revised manuscript.
In the revised version, the sections highlighted in red represent modifications addressing the suggestions from Reviewer 1, while the blue sections reflect changes made in response to Reviewer 2's comments. The teal highlights indicate revisions made in terms of language and writing style.
Furthermore, we would like to show the details as follows:
Point 1: In line 47, There is a typo. "gorup".
Response 1: Regarding this error, in line 61, we have already made revisions.
Round 2
Reviewer 1 Report
The paper is in good shape now after the review.